# Quantitative Imaging and Radiomics in Multiple Myeloma: A Potential Opportunity?

**DOI:** 10.3390/medicina57020094

**Published:** 2021-01-21

**Authors:** Alberto Stefano Tagliafico, Alida Dominietto, Liliana Belgioia, Cristina Campi, Daniela Schenone, Michele Piana

**Affiliations:** 1Department of Health Sciences (DISSAL), University of Genoa, 16129 Genoa, Italy; alida.dominietto@hsanmartino.it (A.D.); liliana.belgioia@unige.it (L.B.); 2IRCCS Ospedale Policlinico San Martino, 16129 Genoa, Italy; 3Department of Mathematics (DIMA), University of Genoa, 16129 Genoa, Italy; campi@dima.unige.it (C.C.); daniela.schenone25@gmail.com (D.S.); piana@dima.unige.it (M.P.); 4CNR—SPIN, 16129 Genoa, Italy

**Keywords:** multiple myeloma, computed tomography, artificial intelligence, radiomics, prognosis, imaging, diagnosis

## Abstract

Multiple Myeloma (MM) is the second most common type of hematological disease and, although it is rare among patients under 40 years of age, its incidence rises in elderly subjects. MM manifestations are usually identified through hyperCalcemia, Renal failure, Anaemia, and lytic Bone lesions (CRAB). In particular, the extent of the bone disease is negatively related to a decreased quality of life in patients and, in general, bone disease in MM increases both morbidity and mortality. The detection of lytic bone lesions on imaging, especially computerized tomography (CT) and Magnetic Resonance Imaging (MRI), is becoming crucial from the clinical viewpoint to separate asymptomatic from symptomatic MM patients and the detection of focal lytic lesions in these imaging data is becoming relevant even when no clinical symptoms are present. Therefore, radiology is pivotal in the staging and accurate management of patients with MM even in early phases of the disease. In this review, we describe the opportunities offered by quantitative imaging and radiomics in multiple myeloma. At the present time there is still high variability in the choice between various imaging methods to study MM patients and high variability in image interpretation with suboptimal agreement among readers even in tertiary centers. Therefore, the potential of medical imaging for patients affected by MM is still to be completely unveiled. In the coming years, new insights to study MM with medical imaging will derive from artificial intelligence (AI) and radiomics usage in different bone lesions and from the wide implementations of quantitative methods to report CT and MRI. Eventually, medical imaging data can be integrated with the patient’s outcomes with the purpose of finding radiological biomarkers for predicting the prognostic flow and therapeutic response of the disease.

## 1. Introduction

Multiple Myeloma (MM) is the second most common type of hematological disease and, although it is rare among patients under 40 years of age, its incidence rises in elderly subjects [1]. This hematological malignant disease is characterized by the autonomous monoclonal proliferation of plasma cells in the bone marrow [2]. More specifically, MM is a cytogenetically heterogeneous disorder of clonal plasma cells in which an excessive production of either monoclonal intact immunoglobulin molecules or immunoglobulin free light chains kappa or lambda is crucial for the development of clinical features [3,4,5,6,7]. Many risk factors for MM are known, including male sex, radiation exposure and monoclonal gammopathy of undetermined significance (MGUS) [1,6,8]. 

MM manifestations are usually identified through hyperCalcemia, Renal failure, Anaemia, and lytic Bone lesions (CRAB). In particular, in MM the normal myeloproliferation is replaced by a pathological one, with important clinical impacts leading to an increasing risk of pathological fractures [2]. This is the main reason why, in 2003, the International Myeloma Working Group (IMWG) replaced the Durie–Salmon system with a revised version (Durie–Salmon system plus), where the diagnostic role of radiography in identifying bone marrow involvement is overtaken by the increased sensitivity of Magnetic Resonance Imaging (MRI) and hybrid Positron Emission Tomography (PET) and Computerized Tomography (CT) data with FDG as tracer (FDG PET/CT) [7,9,10]. Indeed, the extent of the bone disease is negatively related to a decreased quality of life in patients and, in general, bone disease in MM increases both morbidity and mortality [2,4,11,12,13]. Therefore, the detection of lytic bone lesions on imaging, especially CT and MRI, is becoming crucial from the clinical viewpoint to separate asymptomatic from symptomatic MM patients and the detection of focal lytic lesions is becoming relevant even when no clinical symptoms are present [14]. In a recent report, it has been highlighted by Hillengass [15] that bone imaging in MM is important for diagnosis due to the fact that osteolytic lesions are a reason for treatment. Medical imaging is clearly crucial for localization of bone pain and prevention of complications such as pathologic fractures on long bones such as the femur and vertebral collapse. According to Hillengass [15], bone marrow imaging can identify focal lesions which is related to the risk of progression. From the prognostic point of view, it has to be remembered that the presence of a focal lesion detected with imaging is a strong prognostic marker requiring prompt treatment even in the absence of bone destruction. In recent years, the major development acquired in MM imaging is the replacement of standard radiography with CT. Indeed, a study made by the International Myeloma Working Group found lytic lesions using CT on 25% of patients with a negative radiograph [4]. Using MRI, different bone marrow infiltration patterns could be detected: minimal, diffuse, focal, or mixed focal and diffuse. Bone marrow features in MM are clinically relevant because random bone marrow samples of the pelvis may miss a pathologic bone [4]. Detection of osteoporosis in patients with MM using MRI can be useful to separate pathologic fractures from benign osteoporotic fractures, especially using advanced MRI sequences such as diffusion-weighted sequences. PET/CT imaging is both a functional and morphological technique useful for treatment response assessment; indeed, the disappearance of focal lesions is an important prognostic factor well demonstrated in literature. In MM patients with complete remission, if PET/CT detects the presence of focal lesions, the prognosis is worse. On the contrary, patients with a deep remission after treatment and PET negativity have a very good prognosis if combined with minimal residual disease (MRD) negativity. Finally, potentials of CT are also related to the possibility of distinguishing between MM and other metastatic bone lesions. A recent paper by Uygar Mutlu et al. [16] evaluated, with CT, 320 lesions of 207 patients with MM or metastasis using biopsy or clinical examination as ground truth. It was found that high-density areas on CT were more common in metastasis than MM lesions (85.2% versus 19%). Other radiological features differentiated metastasis from MM. These features were: perilesional sclerosis, heterogeneity and ill-defined margins. CT image analysis revealed that high-density areas inside the lesion increased the probability of a metastasis 25-fold [16]. In 2014, the diagnostic criteria for multiple myeloma were updated by international reference organizations. We report here the key point raised by the revised recommendations and consensus [17,18,19]:-Whole-body low-dose CT is recommended by international reference organizations for detection of lytic bone lesions;-Focal myeloma lesions detected on whole-body MRI will indicate symptomatic multiple myeloma requiring therapy;-The IMWG recommends using cross-sectional imaging in the initial work-up: whole-body low-dose CT, MRI, or PET/CT, depending on availability and resources.

18F-FDG-PET/CT could be included in the definition of minimal residual disease after therapy due to its functional capabilities. Wide availability of 18F-FDG-PET/CT is a potential drawback. Nowadays, radiology is pivotal in the staging and accurate management of patients with MM even in early phases of the disease. Medical imaging is used not only to detect bone lesions but also to predict the risk of early progression from smoldering MM (SMM) to active MM and to identify extra-medullary disease [2,12,14]. CT is particularly useful in identifying the sites of either possible pathologic fractures or neurologic complications and to score bone damage quantitatively [7,12]. Further, compared to conventional radiography, PET/CT and whole-body low-dose CT (WBLDCT) are able to detect the presence of active disease in up to 25% to 40% of cases negative at conventional radiography [2]. Following this strictly evidence-based approach, a grade A recommendation has therefore been assigned to the incorporation of these new imaging modalities (WBLDCT and PET/CT) [7,9,10]. 

Yet, the diagnostic and prognostic capabilities of medical imaging in MM are still under investigation and development. In fact, significant variability in image-based prognostic scores is present among different centers and in clinical practice [5,11,13,20,21]. Further, although the updated version of the IMWG criteria accepts the use of CT, WBLDCT and PET/CT to diagnose lytic bone disease in MM, there is still a lack of reliable computational tools for increasing the prognostic value of these modern imaging modalities. In the present review, we will briefly describe the role of new radiological achievements to increase diagnostic potential of medical imaging, with a specific focus on CT. 

## 2. Quantitative Evaluation of Bone CT and Reader’s Experience

As illustrated by a vast amount of scientific literature, daily clinical practice presents a wide usage of CT data for the diagnosis and follow-up of patients with MM, but, at the same time, there is high variability in image interpretation due to different factors [3,4,5,11,12,13,14,22,23]. For instance, it is not always possible to obtain WBLDCT in every patient and often patients with MM receive standard total body CT including thorax and abdominal evaluations. However, after standard reporting of thorax and abdominal findings, in patients affected by MM the focus should be given to small lytic lesions [3,4,5,24]. Further, the largest number of CT examinations of patients affected by MM are performed on the elderly, which implies that multiple degenerative bone changes are likely to influence the radiological report reducing the agreement among readers in CT image interpretation to detect even clinically significant small lytic lesions [6,8,10,11]. As a consequence, it is probably correct to state that quantitative evaluation of MM is already feasible by relying on either the multiparametric imaging approach provided by MRI or the functional one provided by PET, but that this same quantitative assessment based just on CT investigation is still an open issue. As a confirmation of this, an interesting study focused on the comparison of qualitative and quantitative MRI and CT parameters for assessing the involvement of the axial skeleton in MM patients and came to the conclusion that there was no actual benefit from using quantitative parameters for both imaging modalities [25].

As an operational attempt to overcome this deadlock, in order to quantitatively evaluate the status of bone damage, risk of fracture and instability in MM, to reduce reader’s variability, and to assess CT data with good agreement, radiologists and clinicians have developed the Myeloma Spine and Bone Damage Score (MSBDS) [26]. 

The MSBDS scoring system presents several positive aspects:On a series of 70 patients with total body CT available and acquired at the same center, the MSBDS criteria resulted as being fast, reproducible and easy to integrate in daily clinical practice;MSBDS resulted to be useful not only for radiologists specifically trained to assess the musculoskeletal system, but also for clinicians with no formal training in radiology [26];MSBDS correlated well with other quantitative evaluation systems such as the MY-RADS score, supporting the reliability of the MSBDS criteria and suggesting that this scoring system could be reliable for total-body CT in MM patients;MSBDS has the unique feature of being specifically designed and tailored to MM patients while, on the contrary, previously published scoring systems developed mainly in orthopedic environments were designed for spinal assessment in metastatic patients [27];MSBDS not only evaluates the bones to look for spinal instability, but the lytic bony damage is considered a prognostic target. Specific items of MSBDS are dedicated to the proximal femur involvement and to lytic lesions;MSBDS could be far more reliable and diffuse than other scores used for MM patients such as the MY-RADS score and the IMPeTUs criteria for PET or PET/CT [13].At a very practical level, MSDBS can be used on CT images that are far more available than MR images, is very fast and easily reproducible and requires the scoring of a low number of parameters.

The Durie–Salmon System and the International Staging System are still the standard of care in patients with MM for staging. However, the increasing number of patients with MM evaluated with CT needs a thorough evaluation with reliable and quantitative parameters as suggested by recent advancement in precision medicine [28,29]. The MSDBS is ready for immediate clinical application and improves current methods of scoring bone lesions in MM (Table 1). We also point out that MSBDS is only an option to quantitatively score MM bone involvement and it should represent a starting point to correctly evaluate the patient’s impairment not only in clinical practice but also in the medico-legal field (e.g., private health insurances). Every quantitative criteria and potential imaging biomarker must be validated in large trials. At the present time, a prospective clinical validation of MSBDS criteria is underway and results will be available as soon as negative effects of the COVID-19 pandemic are less severe on radiological research [30].

Another important issue in medical imaging related to MM is concerned with the reader’s experience. Radiology is a wide field where different subspecialties are present. For example, the European Society of Radiology, who promotes and coordinates the scientific, philanthropic, intellectual and professional activities of radiology, have several affiliated subspecialties, reflecting great heterogeneity in radiological profession [31]. For MM, the interpretation of bony lesions is straightforward in most cases due to the fact that lesions are lytic. However, in some cases, lesion could be numerous, well circumscribed and could punch out lucencies, raindrop skull, endosteal scalloping and sometimes generalized osteopenia [32]. In these cases, it is likely that a subspecialty with experience in reading CT and MRI data will enhance the role of CT and MRI for MM [5,26]. In many centers, consultation and second-opinion interpretation of medical images by subspecialty radiologists are routinely performed [5,33,34,35,36,37]. A recent study was conducted with the aim of improving the radiological detection and characterization of clinically significant lytic lesions using standard CT (Table 2). Discrepancy rates up to 15% have been reported, describing reports by radiologists at different levels of training and radiologists at different clinical settings, while the discrepancy rate in interpreting a clinically important abnormality (e.g., interpreting the presence of a lytic lesion > 5 mm) reached 21% [5]. A clinical benefit of a subspecialty second-opinion consultation in MM CT has been demonstrated, in particular, for lytic lesions. Indeed, a lytic lesion in MM is sometimes difficult to detect, especially when the diameter is between 5 and 10 mm and when it is located in an osteoporotic and degenerated vertebral body. Particularly for these patients, dedicated musculoskeletal (MSK) radiologists could solve difficult and uncertain cases [5,36,38]. The expertise of a dedicated reader is also crucial because a reference standard is difficult to achieve in MM since bone biopsy cannot be obtained in small lesions and in every anatomical location.

## 3. Radiomics in MM

Radiomics is a very recent approach to the diagnostic, prognostic and therapeutic assessment of specific diseases, which, in its most recent variation, relies on the use of pattern recognition for the extraction of quantitative descriptors from imaging data (typically acquired by means of structural modalities) and on their use with prediction purposes by means of computational algorithms based on Artificial Intelligence (AI) [39,40,41,42,43]. 

Radiomics is performed using dedicated software on standard radiological images. Interestingly enough, a quick search on most databases immediately points out that the number of papers dealing with the use of AI-based radiomics as applied to MM is up to one order of magnitude smaller than the number of papers devoted to the application of this same technique to breast cancer data ([44] and references therein) and even two orders of magnitude smaller than in the case of lung cancer [45] and references therein. The reason for this dramatic difference is probably in the notable genetic heterogeneity of MM, which has notable impacts on the pathogenesis and progression of the disease. In addition, few articles were published in the early 2012/2013 dealing with radiomics and radiology; in the year 2020, it was possible to find more than 1000 articles on PubMed (https://pubmed.ncbi.nlm.nih.gov/?term=%28radiomics%29%29+AND+%28radiology%29&filter=years.2020-2021) reflecting the growing interest in this field. As stated in 2012 by Lambin et al. [40], development of radiomics comes from the potential to overcome some known limitations of bioptical samples in cancer. Indeed, solid cancers, including MM for bone involvement, are characterized by an extraordinarily spatial and temporal heterogeneity involving genes, proteins, cells, microenvironment and tissues. Radiomics could be considered a kind of noninvasive imaging tool able to study and define intratumoural heterogeneity using only data included in medical images and without the need or limitations of a soft-tissue or bone biopsy. From 2012 to 2021, a wide amount of scientific research has been made regarding radiomics and other tools related to the artificial intelligence fields have been adopted. Machine learning and deep learning are strongly associated with radiomics workflow. Machine learning includes computational algorithms using the image features extracted by radiomics as the input in order to provide predictions concerning disease outcomes on follow-up as the output. In MM patients, radiomics and machine learning can be used to predict outcomes on the basis of MRI or CT imaging features. Machine learning can be unsupervised if no information is provided by or determined by an available historical set of data. Machine learning is supervised if machine learning methods are trained with an available data archive. Finally, machine learning with deep learning uses imaging input into a multilayer neural network to sequentially modify the target and reduce its size until a set of numbers is automatically produced.

Indeed, MM is a genetically complex disease that evolves from premalignant stages, such as M-GUS and SMM, and progresses to symptomatic MM [46].

In MM, the development of the disease is very complex and a progression with clonal sweeps in the early phase and a regional evolution in advanced disease have been recently confirmed [6]. Therefore, an analysis of multiple bone lesions could be performed using computer algorithms supporting the use of radiomics. For instance, a recent application of radiomics in MM showed that, in clinical practice, radiomics is able to improve the radiological evaluation of focal and diffuse pattern of MM on CT by improving the Area Under the Curve (AUC) of radiologists [47]. Specifically, accuracy in terms of the AUC of radiologists compared to the reference standard was lower (64%) than accuracy computed using a radiomics approach, which obtained a maximum value of 79%. Further, by using a radiomics approach it is possible to increase the reading accuracy of radiological characterization of focal and diffuse pattern of MM on standard CT [47].

With specific reference to CT data, the use of AI for the assessment of MM radiomics typically relies on the two-step process illustrated in Figure 1:Pattern recognition and property extraction algorithms [21,44] is applied to either specific bone lesions or the whole skeleton asset in order to extract quantitative descriptors of the impact of the disease on the MM bone structure;Machine learning [44,48], in either its unsupervised or supervised version, is applied against the descriptors extracted by step 1 in order to both stratify the MM patients on the basis of their CT data characteristics and predict the disease outcome as far as post-transplantation relapse is concerned.

Preliminary results obtained by means of this kind of approach [48] show that MM is associated with an extension of the intrabone volume for the whole body and that machine learning can identify CT image properties mostly correlating with the disease evolution.

However, the use and validation of radiomics for prognostic purposes in MM is still in progress and several factors have to be considered before assuming radiomics results are completely reliable, repeatable and feasible in clinical practice. In general, medical imaging is capable of generating imaging biomarkers, while acquisition and analysis processes are different from frequently used comparators such as blood or urine biomarkers. This difference is related to the methodology for obtaining the sample. In radiology, the acquisition of the sample (i.e., the data set of images) is heterogeneous by design, since complex equipment from different vendors is clinically used. Even with multiple human and technological efforts, standardization of image quality as an input for different imaging biomarkers analysis is difficult and it is not yet clear whether a complete standardization can be realized. Several scientific societies such as the European Society of Radiology and the Radiological Society of North America (RSNA) tried to provide standards for the best possible standardization at the acquisition level and the minimum requirements for the image analysis software used in the qualification of imaging biomarkers [28,29,41,48,49,50]. However, more sophisticated corrective measures could be taken by artificial intelligence (AI)-based approaches to let complex and deep neural networks learn from the lack of homogeneity in the collected images, both in the DICOM metadata and in the pixel information, and adjust the imaging parameters to be analyzed.

## 4. Conclusions

There is still high variability in the choice between various imaging methods to study MM patients and high variability in image interpretation with suboptimal agreement among readers even in tertiary centers. Therefore, the potential of medical imaging for patients affected by MM is still to be completely unveiled. In the coming years, new insights to study MM with medical imaging will derive from AI and radiomics usage in different bone lesions and from the wide implementations of quantitative methods to report CT and MRI. Eventually, medical imaging data can be integrated with the patient’s outcomes, with the purpose of finding radiological biomarkers for predicting the prognostic flow and therapeutic response of the disease.

## Figures and Tables

**Figure 1 medicina-57-00094-f001:**
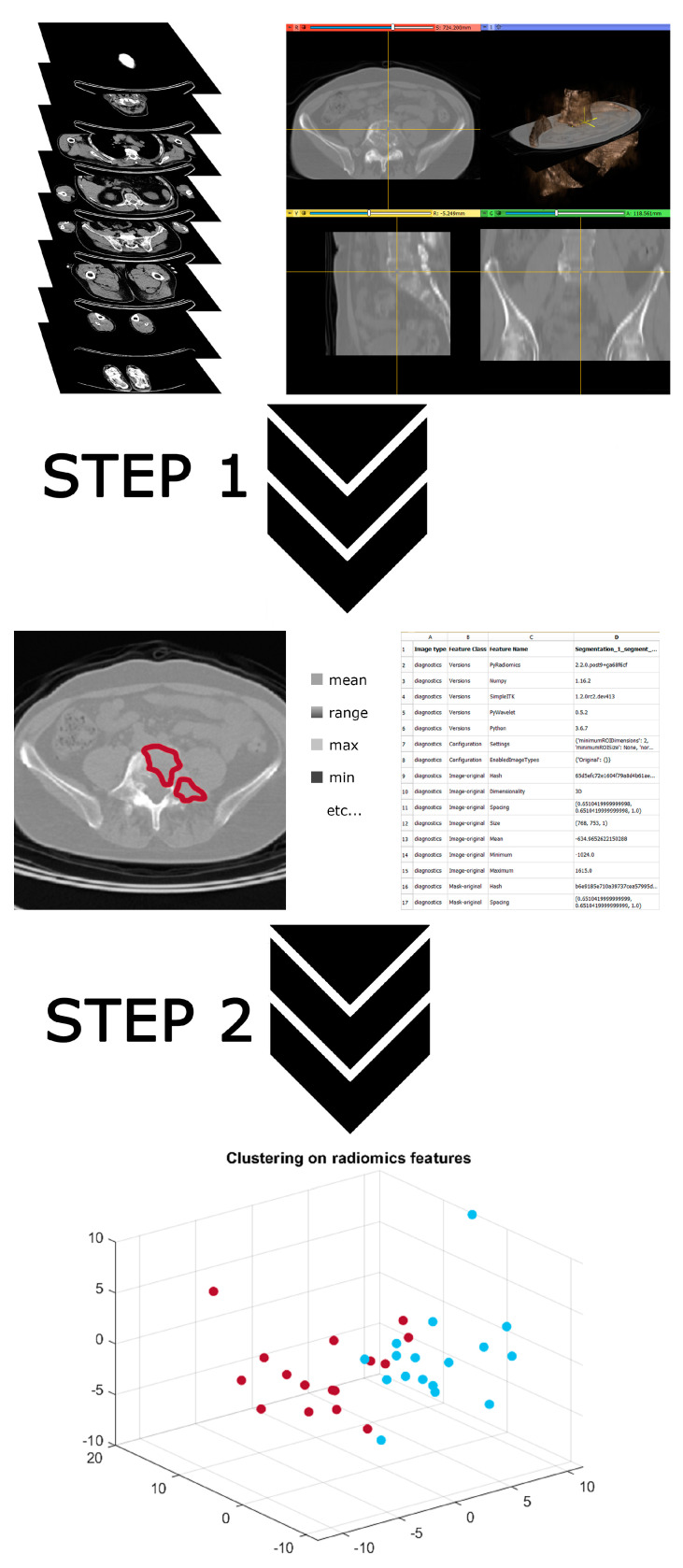
The two-step process of Artificial Intelligence (AI)-based radiomics. The input data (first row) are fed into a property extraction algorithm to obtain a set of image features (second row). These features are fed into a machine learning algorithm to obtain patients’ stratification (third row).

**Table 1 medicina-57-00094-t001:** Myeloma Spine and Bone Damage Score (MSBDS). Interpretation: High risk: >10: immediate surgical or radiation oncologist consultation. Medium risk: ≥5–10: possible instability and medium risk of pathologic fracture. Low risk: <5. * Bone abnormalities not sufficient to give high risk scores, if isolated. ** 1 point for every segment according to MY-RADS (from [26]).

Location	Points
Junctional Spine (C0–C2, C7–T2, T11–L1, L5–S1)	3
Mobile Spine (C3–C6, L2–L4)	2
Collapse/involvement > 50%	3
Collapse < 50% *	2
Posterolateral (facet, pedicle) involvement monolateral	2
Posterolateral (facet, pedicle) bilateral monolateral	3
Spinal Canal involvement	5
Trochanteric region focal lesions < 10 mm	2
Femoral neck or entire trochanteric region	5
More 2/3 of bone diameter	3
Focal lesion > 5 mm at any site *	1
Diffuse Pattern	1 **

**Table 2 medicina-57-00094-t002:** Example of minimal computed tomography technical parameters for lytic lesion detection in multiple myeloma (from: [5]).

Number of Detector Rows	16 or More up to 128
Minimum Scan coverage	Skull base to femur
Tube voltage(kV)/time-current product (mAs)	120/50–70, adjusted as clinically needed
Reconstruction convolution kernel	Sharp, high-frequency (bone) and smooth (soft tissue). Middle-frequency kernel for all images are adjusted by the radiologist as deemed necessary
Iterative reconstruction algorithms	Yes (to reduce image noise and streak artefacts)
Thickness	≤5 mm
Multiplanar Reconstructions (MPRs)	Yes (sagittal, coronal and parallel to long axis of proximal limbs)
Matrix, Rotation time, table speed, pith index	128 × 128, 0.5 s, 24 mm per gantry rotation, 0.8

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
