# Peer review of "Quantitative Imaging and Radiomics in Multiple Myeloma: A Potential Opportunity?"

_medicina, 2021, doi:10.3390/medicina57020094_

Round 1

Reviewer 1 Report

The revised submission has significant improvements, the reviewed topic of CT and computerized radiomics for assessment of MM is quite interesting and helpful for both radiologists and radiomic researchers.

Author Response

English language and style have been checked.

Thank you again for the thorough work of Review.

Reviewer 2 Report

Dear authors

Interesting topic to talk about. If would like to make some comments about your job in order to improve it. First of all, let´s talk about the Introduction. In my opinión, I think it´s a bit out the order. You write first about diagnosis with the different imaging modalities, then prognostic, response, and at the end of the paragraph you talk again about the diagnosis. In my opinion, it is better to understand talking about diagnosis, staging, prognosis, response, and relapse with the different imaging techniques or talk about the "papel" of the different imaging techniques in the different scenarios.

I would like to ask you a question related to MSBDS, is there any publication comparing the accuracy of this score with yIMPeTUs? The bibliography you attached doesn´t compare both techniques (13)

Related to radiomics, could it be possible to explain a little bit deeper how is it performed?

Once again, thank you for your Little review about this topic. Have you got any experience of the new score MSBDS or Radiomics at your Hospital?

Author Response

Interesting topic to talk about. If would like to make some comments about your job in order to improve it. First of all, let´s talk about the Introduction. In my opinión, I think it´s a bit out the order. You write first about diagnosis with the different imaging modalities, then prognostic, response, and at the end of the paragraph you talk again about the diagnosis. In my opinion, it is better to understand talking about diagnosis, staging, prognosis, response, and relapse with the different imaging techniques or talk about the "papel" of the different imaging techniques in the different scenarios.

  • We agree with the Referee, the text underwent several Review cycles and we had to modify it a lot. Since the topics on the introduction are not the main focus of the paper, we did not modify it.

I would like to ask you a question related to MSBDS, is there any publication comparing the accuracy of this score with yIMPeTUs? The bibliography you attached doesn´t compare both techniques (13). Once again, thank you for your Little review about this topic. Have you got any experience of the new score MSBDS or Radiomics at your Hospital?

  • MSBDS is a new score potentially filling a gap in literature. Studies are underway to compare it with IMPeTUs. In our Hospital we are using MSBDS with promising results. Further studies are underway and will be published soon.

Related to radiomics, could it be possible to explain a little bit deeper how is it performed?

  • We added that Radiomics is performed using dedicated software on standard radiological images.

This manuscript is a resubmission of an earlier submission. The following is a list of the peer review reports and author responses from that submission.

Round 1

Reviewer 1 Report

In principle, the auspicious title of the present manuscript deals with an interestingly and presently much discussed topic (quantitative imaging / radiomics in malignant disease). However, there are several major drawbacks which severely limit the impact and quality of the present manuscript.

The English must be revised by a native speaker; there are many typos, missing points etc.

Introduction:

Detection of focal lesions in both CT and MRI: In this context it is never mentioned, that the big advantage of MRI is the possibility to also detect non-osteolytic bone marrow infiltration.

You state: “… and WBLDCT are able to detect the presence of active disease”? How WBLDCT does provide information concerning disease activity / therapy response? This information is usually provided either by PET-CT or MRI, especially DWI.

Quantitative evaluation of bone CT and readers experience:

A major part of this section is a summary of the MSBDS criteria, introduced by the authors of the present manuscript earlier this year. This whole section 2 of the manuscript reads very advertising for the MSBDS criteria instead of providing straight forward overview of current established state-of-the-art studies / knowledge.

The content of Table 2 (which is a (self) citation from reference 5) cannot be retraced by the reviewer. Moreover technical parameters of CT for MM imaging have nothing to do with the current study and must therefore not be show in the context of the present manuscript.

Radiomics in MM:

The first part of the paragraph again sounds very advertising for reference 5 (published by the authors of the present manuscript earlier this year). The second part of the paragraph clearly missed the opportunity to present a overview and summary of current studies of Radiomics in Radiology in general and MM in special.

Citations:

Several citations are inappropriate. E.g. citation 9; Ekert et al. (MRI image data for response assessment…) in the context of WBLDCT / PET-CT.

Some (self) citations are cited incorrectly (e.g. references 19 [Tagliafacio AS et al] and reference 27 [Tagliafacio AS et al].

It remains questionable if some (self)citations (e.g. 37 in the context of figure 1; dealing with radiomics in breast cancer) are mandatory for the present manuscript.

The citations are not uniform; partly incorrect. Must match journals guidline.

Author Response

We would like to thank the Reviewer for the help in improving this manuscript. We followed all suggestions and hope to have improved the manuscript.

In principle, the auspicious title of the present manuscript deals with an interestingly and presently much discussed topic (quantitative imaging / radiomics in malignant disease). However, there are several major drawbacks which severely limit the impact and quality of the present manuscript.

The English must be revised by a native speaker; there are many typos, missing points etc.

- Author's reply: the English language has been professionally edited by our mother-tongue Departmental service.

 Introduction:

Detection of focal lesions in both CT and MRI: In this context it is never mentioned, that the big advantage of MRI is the possibility to also detect non-osteolytic bone marrow infiltration.

- Author's reply: we corrected the phrase and added the big advantage of MRI to also detect non-osteolytic bone marrow infiltration. This big advantage has little evidence to be prognostic relevant so far. We added this consideration in line 57.

From line 57 to line 65 we added a new paragraph dealing with MRI: " In addition, MRI is able to detect non-osteolytic bone marrow infiltration. The potential of MRI in MM is still growing with new data achieved by appliactions such as functional MRI which added to morphological parameters seems correlated to survival. It is recently been published a study on 114 newly diagnosed MM patients with baseline whole-body diffusion-weighted MRI (WB DW-MRI) available that the mean ADC value of bone marrow (L3-S1 and iliac bone) represented an independent risk factor for both progression-free survival and overall survival [15]. Regarding MRI, practical issues are still to be solved: WB-MRI even in MM has extreme variability in the choice of imaging protocols and use of contrast agents. There is a recognized lack standardization of WB-MRI application in clinical practice [16].

We added also two new references:

Zhang L, Wang Q, Wu X, et al (2020) Baseline bone marrow ADC value of diffusion-weighted MRI: a potential independent predictor for progression and death in patients with newly diagnosed multiple myeloma. European Radiology. https://doi.org/10.1007/s00330-020-07295-6

Petralia G, Padhani AR, Pricolo P, et al (2019) Whole-body magnetic resonance imaging (WB-MRI) in oncology: recommendations and key uses. Radiologia Medica. 10.1007/s11547-018-0955-7

You state: “… and WBLDCT are able to detect the presence of active disease”? How WBLDCT does provide information concerning disease activity / therapy response? This information is usually provided either by PET-CT or MRI, especially DWI.

- Author's reply: We apologize for this foggy writing. We corrected the sentence in line 71 and 72 as "Further, compared to conventional radiography, PET/CT, MRI and whole-body low-dose CT (WBLDCT) are able to detect the presence of active or lytic disease earlier."

Quantitative evaluation of bone CT and readers experience:

A major part of this section is a summary of the MSBDS criteria, introduced by the authors of the present manuscript earlier this year. This whole section 2 of the manuscript reads very advertising for the MSBDS criteria instead of providing straight forward overview of current established state-of-the-art studies / knowledge.

- Author's reply: we deeply understand Reviewer 1 opinion. The goal of this manuscript was not to provide straightforward overview of current established state-of-the-art studies / knowledge becuse there are a lot of well written reviews on this topic. Instead, the goal of the present paper is to highlight quantitative imaging and radiomic tools in Multiple Myeloma. We tried to describe the MSBDS criteria because we are still not aware of a strong quantitative tool to report CT in MM. In the next paper we will describe other CT based quantitative tools developed to report bony findings, if present.

The content of Table 2 (which is a (self) citation from reference 5) cannot be retraced by the reviewer. Moreover technical parameters of CT for MM imaging have nothing to do with the current study and must therefore not be show in the context of the present manuscript.

- Author's reply: these are Editorial decisions. We are happy to follow Editor instructions.

Radiomics in MM:

The first part of the paragraph again sounds very advertising for reference 5 (published by the authors of the present manuscript earlier this year). The second part of the paragraph clearly missed the opportunity to present a overview and summary of current studies of Radiomics in Radiology in general and MM in special.

- Author's reply: the goal of this manuscript was not to provide straightforward overview of current established state-of-the-art studies / knowledge becuse there are a lot of well written reviews on this topic. In addition, the present paper is not an overview of current studies of Radiomics in Radiology (there are many) in general. If we use on pubmed the keyword:" Radiomics and Multiple Myeloma" there are less that 5 papers, the majority non pertinent. The first ever published is "Tagliafico AS, Cea M, Rossi F, Valdora F, Bignotti B, Succio G, Gualco S, Conte A, Dominietto A. Differentiating diffuse from focal pattern on Computed Tomography in multiple myeloma: Added value of a Radiomics approach. Eur J Radiol. 2019 Dec;121:108739. doi: 10.1016/j.ejrad.2019.108739. Epub 2019 Nov 7. PMID: 31733431." The others are cited, if pertinent. We added new considerations on line 208-212.

Citations:

Several citations are inappropriate. E.g. citation 9; Ekert et al. (MRI image data for response assessment…) in the context of WBLDCT / PET-CT.

Some (self) citations are cited incorrectly (e.g. references 19 [Tagliafacio AS et al] and reference 27 [Tagliafacio AS et al].

It remains questionable if some (self)citations (e.g. 37 in the context of figure 1; dealing with radiomics in breast cancer) are mandatory for the present manuscript.

The citations are not uniform; partly incorrect. Must match journals guidline.

- Author's reply: citations have been done with software, Mendeley. We apologize for inaccuracy and we checked them also manually.

Reviewer 2 Report

A well written review paper. The introduction has adequate references to the MM diseases regarding risk factors, and the roles of imaging. As mentioned, the CT, MRI and PET have been increasingly  used in the clinical setting for different task such as detection, diagnosis, treatment assessment, although this paper mainly focused on the CT, the title has a little misleading that the "quantitative imaging" covers wide spectrum of imaging modalities of MRI, PET, X-ray, etc.  The section 2 of CT and reader's experience seems mainly described MSBDS which was based on 70 patients that may not support strong evidences and has some misleadings, especially it could be far w=more reliable and diffuse than other scores. Regarding the sensitivity of lesion detection, the MRI and PET could be better, especially the assessment of bone marrow. For radiomics been used in machine learning, there is a wide variety of radiomics reported in the literatures, and very commonly, the effective radiomics have to be selected from the large amount of the radiomics extracted from the CT images using the training set, in which the small training set may limit the generalization of the radiomics in patient population. Unfortunately, most studies reported the performance based on the small data set both in training and test. Nonetheless, the radiomic has potentials and limitations that need to be discussed in this review paper.

Author Response

We would like to thank the Reviewer for the help in improving this manuscript. We followed all suggestions and hope to have improved the manuscript.

A well written review paper. The introduction has adequate references to the MM diseases regarding risk factors, and the roles of imaging. As mentioned, the CT, MRI and PET have been increasingly  used in the clinical setting for different task such as detection, diagnosis, treatment assessment, although this paper mainly focused on the CT, the title has a little misleading that the "quantitative imaging" covers wide spectrum of imaging modalities of MRI, PET, X-ray, etc.  The section 2 of CT and reader's experience seems mainly described MSBDS which was based on 70 patients that may not support strong evidences and has some misleadings, especially it could be far more reliable and diffuse than other scores. Regarding the sensitivity of lesion detection, the MRI and PET could be better, especially the assessment of bone marrow. For radiomics been used in machine learning, there is a wide variety of radiomics reported in the literatures, and very commonly, the effective radiomics have to be selected from the large amount of the radiomics extracted from the CT images using the training set, in which the small training set may limit the generalization of the radiomics in patient population. Unfortunately, most studies reported the performance based on the small data set both in training and test. Nonetheless, the radiomic has potentials and limitations that need to be discussed in this review paper.

- Author's reply: From line 57 to line 65 we added a new paragraph dealing with MRI: " In addition, MRI is able to detect non-osteolytic bone marrow infiltration. The potential of MRI in MM is still growing with new data achieved by appliactions such as functional MRI which added to morphological parameters seems correlated to survival. It is recently been published a study on 114 newly diagnosed MM patients with baseline whole-body diffusion-weighted MRI (WB DW-MRI) available that the mean ADC value of bone marrow (L3-S1 and iliac bone) represented an independent risk factor for both progression-free survival and overall survival [15]. Regarding MRI, practical issues are still to be solved: WB-MRI even in MM has extreme variability in the choice of imaging protocols and use of contrast agents. There is a recognized lack standardization of WB-MRI application in clinical practice [16].

We added also two new references:

Zhang L, Wang Q, Wu X, et al (2020) Baseline bone marrow ADC value of diffusion-weighted MRI: a potential independent predictor for progression and death in patients with newly diagnosed multiple myeloma. European Radiology. https://doi.org/10.1007/s00330-020-07295-6

Petralia G, Padhani AR, Pricolo P, et al (2019) Whole-body magnetic resonance imaging (WB-MRI) in oncology: recommendations and key uses. Radiologia Medica. 10.1007/s11547-018-0955-7

We added new considerations on line 208-212.

Radiomic has potentials and limitations are discussed further in this review paper after line 238.

Round 2

Reviewer 1 Report

After (minor) Revision there are still several major drawbacks which severely limit the impact and quality of the present manuscript. Unfortunately the authors clearly missed the opportunity to present an deeply revised manuscript; instead they added a few paragraphs without changing the gerneral tenor of the manuscript. The fact that several / major parts of the manuscript are advertising self citations and summary’s of recently published other papers of the author (which do not contribute to the actual idea of the manuscript) has not been adressed.